# Investigating Neural Reward Sensitivity in the School Grade Incentive Delay Task and Its Relation to Academic Buoyancy

**DOI:** 10.3390/bs15101321

**Published:** 2025-09-26

**Authors:** Myrthe J. B. Vel Tromp, Hilde M. Huizenga, Brenda R. J. Jansen, Anna C. K. van Duijvenvoorde, Ilya M. Veer

**Affiliations:** 1Department of Developmental Psychology, Faculty of Social and Behavioural Sciences, University of Amsterdam, Nieuwe Achtergracht 129, 1018 VZ Amsterdam, The Netherlands; h.m.huizenga@uva.nl (H.M.H.); b.r.j.jansen@uva.nl (B.R.J.J.); i.m.veer@uva.nl (I.M.V.); 2Department of Developmental and Educational Psychology, Faculty of Social and Behavioural Sciences, Leiden University, Wassenaarseweg 52, 2333 AK Leiden, The Netherlands; a.c.k.van.duijvenvoorde@fsw.leidenuniv.nl

**Keywords:** academic buoyancy, reward anticipation, School Grade Incentive Delay task, ventral striatum

## Abstract

Understanding the mechanisms behind academic buoyancy, the ability to effectively cope with everyday academic challenges, is essential for identifying the factors and mechanisms that help students maintain their motivation and cope with routine academic pressures. One potential underlying mechanism is reward sensitivity, or the capacity to experience pleasure both in anticipating and receiving reward-related stimuli. We hypothesized that individuals with higher sensitivity to anticipated reward would exhibit greater academic buoyancy. To test this in an academic context, we modified the Monetary Incentive Delay (MID) task into a School Grade Incentive Delay (SGID) task, where participants work towards a fictitious school grade by winning or losing points on each of the trials. In this study, we investigated whether the SGID activates the neural reward circuitry similar to the traditional MID and whether this is associated with academic buoyancy. The SGID task activated key brain regions associated with reward anticipation, validating its use for studying reward processing in academic contexts. Importantly, we found a negative association between academic buoyancy and right amygdala activation during reward anticipation, suggesting that buoyant students may benefit from reduced emotional reactivity when anticipating rewards. Further research in larger samples is needed to capture the full complexity of reward processing in relation to academic buoyancy.

## 1. Introduction

Understanding why students remain motivated to achieve good academic outcomes despite challenges at home or school is a key question in educational psychology. This phenomenon is often described in two related but distinct ways: academic resilience, which refers to positive adaptation in the face of chronic or major adversities, and academic buoyancy, which refers to students’ ability to successfully deal with the more common, everyday academic setbacks and pressures ([28]). While only a minority of students face chronic adversities, almost all students face daily academic hassles ([30]). Because not all students are equally affected by these hassles, research should focus on identifying factors and mechanisms that could explain why some students adapt more easily than others ([19]), in order to help students maintain their motivation and cope with routine academic pressures.

One promising factor is reward sensitivity, or the capacity to experience pleasure both in anticipating and receiving reward-related stimuli ([17]). At the self-report and behavioral level, reduced reward sensitivity is impaired in disorders such as posttraumatic stress disorder ([2]; [12]; [21]) and major depressive disorder ([23]; [34]). Not surprisingly, anhedonia, the diminished capacity to feel enjoyment or reward satisfaction, is commonly found in depressive disorders ([31]). Stressors may play a significant role in the onset of anhedonic symptoms and associated mental disorders ([8]). Research suggests that individuals with higher reward sensitivity display higher levels of positive affect following stressor exposure ([7]) and may better maintain their mental health, compared to individuals with lower reward sensitivity ([16]). These findings suggest that behavioral reward sensitivity can buffer against the negative effects of stress.

At the neural level, reward sensitivity refers to the responsiveness of fronto-striatal and limbic circuits, particularly the ventral striatum, nucleus accumbens, and amygdala, during reward anticipation and outcome ([36]). Studies converge on the idea that reward processing supports resilience. For instance, stronger nucleus accumbens (Nacc) activation in response to a reward task predicts fewer depressive symptoms following stress exposure ([39]), and heightened ventral striatal reward reactivity has been associated with decreases in depressive symptoms over time ([40]). During reward anticipation specifically, amygdala and Nacc activation moderated the association between the number of life stressors and depressive symptoms ([14]). These findings are consistent with the reward prediction error model, where dopaminergic neurons code deviations between expected and actual outcomes ([36]), and with reinforcement learning research showing that striatal activity tracks prediction errors across paradigms ([15]). Reward sensitivity supports adaptive decision-making under uncertainty, shaped by individual differences in reinforcement learning parameters ([10]; [6]). Importantly, acute stress can dampen reward sensitivity, reducing striatal and orbitofrontal responses to rewards ([35]), and psychosocial stress reduces neural reward responsivity ([13]). Together, this suggests that neural sensitivity to reward anticipation may be particularly important for resilient outcomes ([41]).

Although it has been suggested that individual differences in responses to both the anticipation and the receipt of reward may be important mechanisms underlying resilience ([11]), there is evidence to suggest that specifically the anticipation phase is important ([41]). For example, stress has been shown to potentiate motivation and “wanting” during the anticipatory phase but reduce reward responsiveness during the consummatory phase ([26]). Importantly, recent neuroimaging work indicates that activation in the bilateral amygdala and the Nacc during reward processing can buffer the impact of life stressors on depressive symptoms ([14]). Notably, this buffering effect was observed during the anticipation of rewards, not during the receipt of rewards, highlighting the particular relevance of reward anticipation in achieving a resilient outcome. In the context of academic stressors, this may explain why some students remain motivated and engaged despite encountering daily academic hassles. Accordingly, we hypothesize that individual differences in reward sensitivity, specifically during the anticipation of rewards, may play a key role in students’ academic buoyancy.

Reward sensitivity can be measured at the neural level using the Monetary Incentive Delay (MID) task ([22]). The task differentiates between the neural response to (a) the anticipation of gain or loss and (b) the outcome where participants receive said monetary gain or loss ([24]). This task triggers robust neural responses during the anticipation of monetary gains and losses ([45]), including in the striatum, insula, amygdala, and thalamus ([32]), which are core brain areas for reward processing ([27]). Moreover, during the receipt of reward, the task activates the medial orbitofrontal cortex (mOFC) and dorsal anterior cingulate cortex (dACC) for both gains and losses ([5]). However, in an academic context, the main source of reward comes from school grades, and these grades act as an important source of external motivation for students ([25]; [38]). Therefore, to obtain a more ecologically valid measure of reward in educational contexts, we modified the MID task to a School Grade Incentive Delay (SGID) task.

In the MID, participants can win or lose money in every trial and build toward a final amount of money at the end of the task. In the SGID, participants can win or lose points that are added or subtracted towards a final grade which is given at the end of the task. One major difference between the two tasks is that the money is paid out at the end of the MID, while the grade obtained in the SGID does not affect the participant’s actual academic career. These differing real-life consequences may influence the (strength of) neural responses, so it is important to examine the validity of the SGID. We hypothesized that the neural activations observed during the SGID would be similar to the activations evoked by the MID for gain and avoiding loss in both the anticipation and outcome phase. Beyond this validation, we also explored individual differences in neural responses. We expected that individuals who display a stronger neural response to the anticipation of rewards would be more academically buoyant. Finally, as regions associated with win and loss anticipation largely overlap ([5]), we tested whether stronger neural responses to the anticipation of losses also were associated with academic buoyancy.

## 2. Materials and Methods

### 2.1. Participants

The present study was conducted as a pilot study for a larger study embedded in the Growing Up Together in Society (GUTS) consortium (https://gutsproject.com; see [9] for an overview paper). We aimed to recruit 30 participants to obtain a solid estimate of SGID activation, as comparable studies found robust activation on the MID with similar sample sizes (e.g., [3]). Participants were recruited through online participant pool management systems (SONA systems; https://sona-systems.com) of the Vrije Universiteit Amsterdam and the Spinoza Centre for Neuroimaging, Amsterdam, The Netherlands. To participate in the study, participants had to be between 18 and 30 years old, speak Dutch fluently, have no MRI contraindications, have normal or corrected-to-normal vision, and not have a (history of) psychiatric illnesses. In total, we recruited 37 participants (23 female, 12 male, 1 non-binary, 1 unknown), with a mean age of 23.64 (range = 19–29, sd = 2.46). All participants were currently enrolled as a higher education student or completed a higher education degree (higher vocational education (N = 3) or university (N = 33); one unknown higher education). Participants signed informed consent prior to inclusion. This study was approved by the institutional review board of the Faculty of Behavioural and Movement Sciences of the Vrije Universiteit Amsterdam (VCWE) under file number VCWE-2023-114.

### 2.2. Measures

#### 2.2.1. School Grade Incentive Delay Task

We modified the MID used in two recent resilience studies (MARP: [20]; DynaMORE: [42]) to evoke neural responses of anticipating a potential gain or loss of points towards a fictitious school grade, and neural responses during the outcome phase. In the outcome phase, participants received feedback on whether they obtained points or managed to avert losing points. We followed the Dutch academic system, in which students receive a grade between 1 and 10 (1 = very poor, 10 = excellent). Participants started the task with an initial grade of 5, which is deemed ‘just insufficient’, to allow most students to obtain a final grade between 7 and 9 (deemed sufficient to good).

The task was programmed and presented using Presentation (Neurobehavioralsystems; https://neurobs.com). Participants were first presented with a fixation cross (jittered duration between 2 and 6 s), followed by the target cue (2 s). These cues encompassed five conditions, with 9 trials per condition: a substantial gain (+1 point), a minor gain (+0.5 point), a neutral condition (no gains or losses), a minor loss (−0.5 point), and a substantial loss (−1 point), mirroring the MID task used by [20] ([20]). The gain cues were accompanied by a circular shape, the loss cues by a square shape, and the neutral condition by a diamond shape. After a second delay marked by another fixation cross (jittered duration between 2 and 2.5 s), participants were presented with a white star (the target) and required to respond with a button press as fast as possible. The target duration varied from 150 to 500 ms based on previous task performance, which was adapted to obtain a 66% hit rate (explained in more detail below). Finally, participants were presented with feedback (2 s) regarding the success of their response, with green text representing a timely button press and red text indicating a response that took too long, as well as a mention of the points gained or subtracted on that trial and the current ‘final grade’ (see Figure 1 for a graphic representation of trials). Pressing the button in time resulted in obtaining the reward in the gain condition, averting loss in the loss condition, and no gains or losses in the neutral condition. Pressing the button too late or not at all resulted in a missed reward in the gain condition, losing the points in the loss condition, and no gains or losses in the neutral condition. As in [20] ([20]), we used 9 trials per condition, leading to a total of 45 trials. We used two quasi-randomized trial orders, in which a maximum of two consecutive trials with the same condition was allowed.

To ensure that the reward experience remained consistent across participants, and consistently with earlier MID studies (e.g., [20]), we aimed to achieve a 66% hit rate within participants and conditions. To this end, an adaptive algorithm that dynamically adjusted the target duration for each participant was used, incorporating performance on the previous trials. If the participant’s hit rate in the past three trials of a certain condition dropped below 66%, the target duration was increased by 25 ms, whereas if the hit rate exceeded 66%, the target duration was decreased by 25 ms.

#### 2.2.2. Academic Buoyancy Scale

We measured academic buoyancy using the Academic Buoyancy Scale (ABS; [28]). The ABS consists of four items (e.g., “I’m good at dealing with setbacks at school, such as negative feedback on my work or a poor result”) that were translated into Dutch by the first author. Participants could indicate how much they agreed with the statement on a seven-point Likert scale where 1 indicated “strongly disagree” and 7 indicated “strongly agree”, yielding a sum score between 4 and 28 with higher scores indicating stronger buoyancy. Previous studies have demonstrated that the ABS is validated, unidimensional, consistent across key sociodemographic factors (such as age, ethnicity, and gender), roughly follows a normal distribution, and is reliable ([28], [29]). In the present study, Cronbach’s alpha was acceptable (α = 0.74).

### 2.3. Procedure

Upon arrival at the research facility, the participant provided informed consent and filled out the MRI contraindication form, which was reviewed by the MRI certified researcher present. Next, we explained the tasks and provided information on how an MRI scanner operates and asked participants to fill in a short questionnaire relevant to another task not described in this paper. Participants were then positioned in the MRI scanner for testing, which took roughly 45 min. We obtained a structural scan, fMRI data of a task unrelated to this paper, and fMRI data of the SGID task. After the scan session, participants filled out an exit questionnaire on an iPad, including measures of task engagement and fatigue as well as the Academic Buoyancy Scale. In this exit questionnaire, participants could indicate whether they found the task (a) enjoyable and (b) convincing, that is, that the task felt like they were achieving a ‘real’ school grade, in two open questions. Participants were rewarded 25 euros or research credits for participation and received reimbursement for travel costs.

### 2.4. Data Acquisition

#### 2.4.1. Behavioral Data

We collected response data (hits, misses, reaction times) via MRI-compatible button boxes (Current Designs; https://www.curdes.com) during the SGID.

#### 2.4.2. MRI Data

Data were collected on a 3.0 Tesla Philips Ingenia CX MRI scanner, using a 32-channel-phased array head coil (Philips Medical Systems, Best, The Netherlands). We utilized foam pads to limit head movement and instructed participants to keep as still as possible. Visual stimuli were presented on a screen behind the scanner bore, which the participant could see through a mirror attached to the head coil. Since the SGID is self-paced, the duration of the task differed slightly per participant (between 243 and 247 volumes). We used a two-dimensional echo planar imaging-sensitivity encoding sequence, with a 3 mm isotropic voxel size and the following parameters: repetition time (TR): 2000 ms; echo time (TE): 27.63 ms; flip angle = 76.1°; field of view 240 × 240; 37 slices with a 0.3 mm gap. We additionally acquired a T1-weighted structural image for registration purposes (voxel size = 1 mm isotropic, TR = 8.2 ms, TE = 3.7 ms, flip angle = 8°, field of view 240 × 188; 220 slices).

### 2.5. Data Analysis

MRI data. MRI data were visually inspected for artifacts (none were found) and pre-processed and analyzed using FSL ([18]). Preprocessing steps included skull stripping, motion correction, spatial smoothing (FWHM = 6 mm), grand mean scaling by a single multiplicative factor, and Gaussian high-pass temporal filtering (cutoff of 125 s). Functional images were first co-registered to the participant’s T1-weighted image and then normalized to the MNI-152 standard space template.

In the first level analysis we included 17 explanatory variables (EVs), comprising the following events: Five EVs for the anticipation period ([1] no win or loss, [2] low or [3] high gain, and [4] low or [5] high loss) spanning cue offset to target onset, ten EVs for the feedback period (hits and misses for [6,7] no win or loss, [8,9] low or [10,11] high win, and [12,13] low or [14,15] high loss) spanning the two seconds after feedback onset, and two EVs for [16] onsets of all cues (two-second duration) and for [17] all targets (duration equal to participant’s reaction times). Additionally, the six rigid-body motion parameters estimated during preprocessing were added as EVs of no interest to account for head motion in the analysis. Each EV was convolved with a double gamma hemodynamic response function. Next, primary contrasts of interest were specified: anticipation of a win > anticipation of no win or loss (“Anticipation: Gain > Zero”), anticipation of a loss > anticipation of no win or loss (“Anticipation: Loss > Zero”), feedback on hits on all gain trials > feedback on misses on all gain trials (“Outcome: Gain > No Gain”), and feedback on hits on loss trials > feedback on misses on loss trials (“Outcome: No Loss > Loss”).

For the second-level analyses, we performed permutation tests on the individual z-values of the contrasts of interest using FSL’s Randomise ([44]), with 1000 random samples to generate a null distribution against which the true finding was tested. First, the main task effect (intercept) across all participants was assessed for each contrast of interest. The significance threshold was set at *p* < 0.05, Threshold-Free-Cluster-Enhancement-corrected for multiple comparisons (TFCE; [37]). Next, associations with the ABS were assessed for the contrast “Anticipation: Gain > Zero”, and exploratively for the contrast “Anticipation: Loss > Zero”. Here, we applied a small-volume correction, separately for three regions of interest (ROIs) that were identified as relevant for this brain-behavior association: bilateral Nacc, bilateral amygdala, and ventromedial PFC (vmPFC). The ROI masks were independently created using the Harvard-Oxford cortical and subcortical atlases, using the versions with a 25% probability threshold in FSL. A Bonferroni correction was applied for testing three separate ROIs, thus setting the significance threshold at *p* < 0.0167, TFCE-corrected for multiple comparisons.

## 3. Results

### 3.1. Descriptive Statistics

Two participants were excluded from the analyses: One participant did not understand the task correctly and therefore had no correct trials, while the other participant admitted ‘cheating’ on the task and therefore obtained an artificially high final grade. The final sample thus consisted of 35 participants (23 female, 10 male, 1 non-binary, 1 unknown) with a mean age of 24.13 (SD = 2.54). Of these participants, three did not fill out the ABS and were thus excluded from analyses pertaining to the ABS.

We calculated descriptive statistics for the Academic Buoyancy Scale (ABS). An average sum score of 16.88 was found (SD = 4.62, range: 7–25), indicating moderate levels of academic buoyancy in the sample. The distribution was examined for normality: skewness = −0.35 and kurtosis = 2.38, indicating a nearly normal distribution with a slight negative skew and marginally lighter tails.

### 3.2. Behavioral Findings

The SGID algorithm aims for a target hit rate of 66%, which was roughly achieved with an observed average 63.5% hit rate (SD = 4.7%; range = 51.11–71.11%). A repeated-measures ANOVA with incentive magnitude as within-subject factor revealed a small but significant effect (*p* < 0.001), with post hoc tests indicating differences between 0 and +0.5 as well as between –0.5 and –1, but including RTs in the fMRI analyses did not affect the results. The mean final grade was 8.8 (SD = 1.76, range = 3.5–12). Notably, the final grade was higher than 10 for eight participants, which is impossible in the Dutch school system and thus a limitation in the present study. Most participants thought that the task was both fun and somewhat frustrating, based on responses to the open questions. The task also appeared to feel convincing, with 70% of participants indicating that the school grade paradigm felt ‘real’ and that they aimed to ‘achieve a good grade’.

### 3.3. fMRI Findings

The fMRI results described in this article are available on Neurovault (see https://neurovault.org/collections/ZVTIRUHL). Please find an overview of all local maxima in Appendix A.

#### 3.3.1. Anticipation Phase

The SGID evoked significant activation related to reward anticipation in the bilateral NAcc and the anterior insula (AI) for both anticipation contrasts, consistent with several meta-analyses of reward processing ([32]; [5]) and the original MID task by [20] ([20]). Additionally, as seen in panel A and B in Figure 2, the results of the “Anticipation of Gain > Zero” contrast closely match activations from a meta-analysis of roughly 7000 participants on reward tasks ([43]), including in the precuneus and the insula. While not as strong, the “Anticipation of Loss > Zero” contrast also engages areas found in the Waller et al. meta-analysis, including in the left hippocampus (see Figure 2, panel C and D). These results indicate that our task successfully evokes activation in brain regions associated with anticipatory reward processing, as observed in traditional MID tasks.

To investigate the incentive magnitude manipulation (1 vs. 0.5 points), we examined activation across the five incentive magnitudes separately for left and right nucleus accumbens (NAcc). Figure 3 displays the mean activation (with SEM) for each magnitude. A clear parametric modulation of incentive magnitude was observed in both left and right NAcc activation, consistent with previous findings using the MID task ([22]; [4]) and with [20] ([20]) employing the same task with monetary rewards.

Post hoc contrasts did not reveal significant differences between anticipation of high versus low reward in either hemisphere (both *p* > 0.1), nor between high versus low loss in the left (*p* = 0.0153). In the right NAcc, however, activation was greater during anticipation of a high compared to a low loss (*p* = 0.01).

#### 3.3.2. Feedback Phase

Consistent with the original MID task by [20] ([20]), we found activation in the bilateral nucleus accumbens for the Gain > No Gain contrast. Additionally, as seen in panel A and B in Figure 4, the results of the Gain > No gain contrast overlap with the [43] ([43]) meta-analysis. In contrast to the original task by [20] ([20]) and our expectations, we did not find activation in the nucleus accumbens for the No loss > Loss contrasts (see Figure 4). Parts of our activation appear to match the Waller meta-analysis, although we found less activation (see panel C and D in Figure 4).

#### 3.3.3. Association Between Academic Buoyancy and Anticipation of Reward and Loss

We found a negative association between academic buoyancy and activation in the right amygdala when anticipating reward (see Figure 5), which was not found when anticipating loss. No associations with academic buoyancy were found for either contrast within the Nacc or vmPFC ROIs.

## 4. Discussion

In this study, we modified the traditional Monetary Incentive Delay (MID) task to a School Grade Incentive Delay (SGID) task, aiming to provide a more ecologically valid measure of reward sensitivity in academic contexts and to test the association between reward sensitivity and academic buoyancy. We found that the SGID task activated brain regions associated with reward anticipation, including the bilateral nucleus accumbens and anterior insula, similar to findings from previous MID studies ([1]; [20]) and a large meta-analysis of neural reward anticipation ([43]). Moreover, nucleus accumbens activation was also found when receiving a reward during the feedback phase, which is in concurrence with previous results ([20]; [45]). In contrast, we did not find activation of the nucleus accumbens or other reward-related brain regions for having averted a loss. As we are specifically interested in reward sensitivity, we can conclude that our task effectively evokes brain activation related to reward processing and can therefore be used to assess reward sensitivity in an educational context in an ecologically more valid way.

Beyond validating the SGID, we examined associations between neural activation and academic buoyancy. After applying small-volume correction within our a priori regions of interest (bilateral Nacc, bilateral amygdala, and vmPFC), we found a negative association between academic buoyancy and activation in the right amygdala during reward anticipation. Contrary to our expectations, we did not find an association between academic buoyancy and ventral striatal and vmPFC activation.

These results appear to diverge from resilience research showing that stronger amygdala activation during reward anticipation buffered against the impact of life stressors ([14]). Other studies similarly highlight that stress reduces neural reward responsibility, while stronger baseline reward responsiveness predicts lower stress reactivity ([33]). Together, these findings support the idea that reward sensitivity plays a protective role under conditions of academic or emotional strain. Tentatively, the negative correlation between academic buoyancy and amygdala activation during reward anticipation may point to an alternative mechanism. Rather than heightened emotional responses, buoyant students may benefit from regulating affective reactivity to maintain focus and motivation under pressure. Given the exploratory nature of this finding and its divergence from prior work, further research is needed to clarify the role of the amygdala in academic stress contexts.

In conclusion, our findings suggest that the SGID task effectively activates brain regions implicated in reward processing that are observed in MID tasks, providing a promising tool for examining reward sensitivity in academic contexts. Our finding of a negative association between academic buoyancy and right amygdala activation during reward anticipation provides a first indication that buoyancy may be linked to affective downregulation rather than heightened reactivity and warrants further investigation. As part of the GUTS consortium ([9]), we are presently conducting a large-scale study in which we administer the SGID in a large sample of 10–14-year-old children and assess both trait and state measures of academic buoyancy to probe its dynamic, fluctuating nature. In this manner, we aim to provide clearer insights into how reward sensitivity supports students’ buoyancy in the face of everyday academic pressures.

## Figures and Tables

**Figure 1 behavsci-15-01321-f001:**
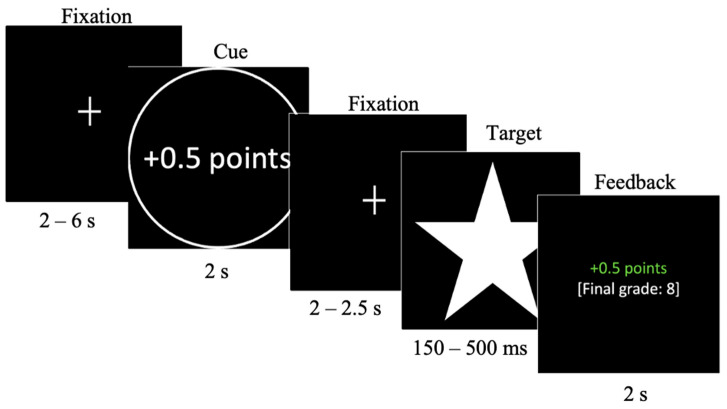
Structure of a gain trial during the SGID task.

**Figure 2 behavsci-15-01321-f002:**
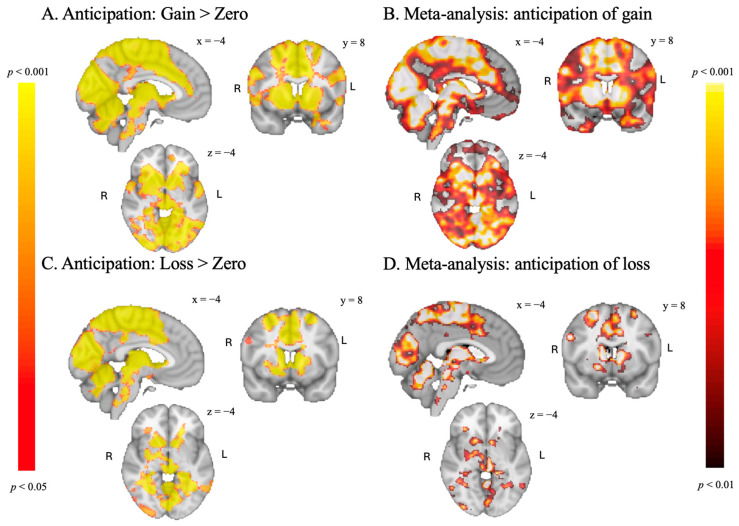
(**A**,**C**): results of the anticipation phase of the SGID, TFCE-corrected at *p* < 0.05. (**B**,**D**): results of a reward task meta-analysis ([43]), *p* < 0.01, uncorrected.

**Figure 3 behavsci-15-01321-f003:**
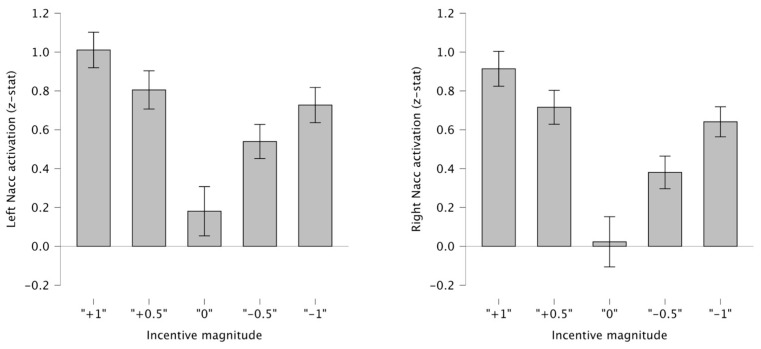
Parametric modulation of nucleus accumbens activation by incentive magnitude during reward anticipation. Bars represent mean activation (with SEM) for the five incentive magnitudes (−1, −0.5, 0, +0.5, +1 points).

**Figure 4 behavsci-15-01321-f004:**
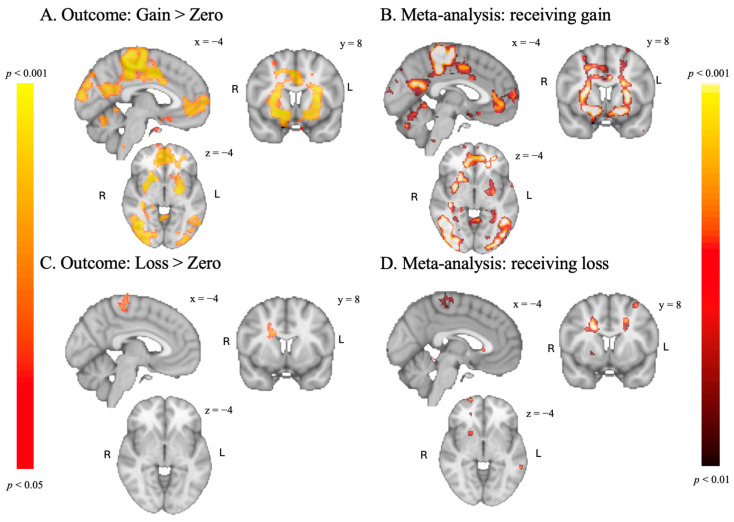
(**A**,**C**): results of the feedback phase of the SGID, TFCE-corrected at *p* < 0.05. (**B**,**D**): results of a reward task meta-analysis ([43]), *p* < 0.01, uncorrected.

**Figure 5 behavsci-15-01321-f005:**
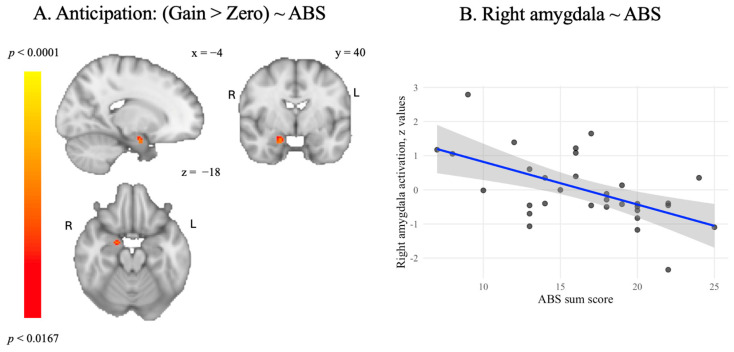
(**A**): results of the association between ABS scores and activation for the anticipation of reward, small-volume TFCE-corrected at *p* < 0.0167. (**B**): scatterplot illustrating the association (blue line) and 95% confidence interval (grey area) between ABS scores and right amygdala activity when anticipating a re-ward.

## Data Availability

The fMRI results described in this article are available on Neurovault (see https://neurovault.org/collections/ZVTIRUHL).

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
