# Peer review of "Investigating Neural Reward Sensitivity in the School Grade Incentive Delay Task and Its Relation to Academic Buoyancy"

_behavsci, 2025, doi:10.3390/bs15101321_

Round 1
Reviewer 1 Report
Comments and Suggestions for Authors
Review Report for The Manuscript behavsci-3711664
General Comments
Dear Editors,
I have thoroughly reviewed the manuscript titled “Investigating Neural Reward Sensitivity in the School Grade Incentive Delay Task and its Relation to Academic Buoyancy”.
This study investigates the neural (fMRI) correlates of reward sensitivity in academic contexts and its association with academic buoyancy. The authors modified the canonical MID task into SGID paradigm. The SGID successfully activated some reward-related regions, validating their neural engagement. However, the authors observed no significant association between academic buoyancy and neural reward-related activation, indicating the need for a more comprehensive and dynamic assessment. While the small sample size and static measurement of buoyancy limit statistical power and inter-individual variability, especially regarding striatal activation, the authors appropriately partially acknowledge these limitations and other unexpected results and discuss them constructively. This provides a solid basis for a forthcoming large-scale implementation, but I would like it to be elaborated further. I enjoyed reading the manuscript which is clearly presented, but at this stage I think some work is still needed before the manuscript can be ready for publication, and I would appreciate it if the authors could consider a few adjustments.
In particular, I recommend that the authors acknowledge more of the relevance of other literature exploring the basis of reward expectation, sensitivity, and modulation across behavioral, neural, and computational domains, which is currently lacking in their manuscript. I believe these changes would strengthen the work and avoid overlooking significant contributions in the field. Addressing these concerns will improve the manuscript’s contextualization and accuracy. Such modifications should not require substantial effort, and I trust the author will take this step to strengthen the quality of their work.
These points are more specifically detailed below.
Detailed Comments
To strengthen the theoretical and neuroscientific framing of the paper and its relation to literature on the neural and behavioral correlates of reward sensitivity and anticipation, I recommend the authors consider integrating some key findings from the literature that are currently missing. While most of the core studies employing the MID task and fMRI meta-analyses are appropriately cited, some important references (some of them pivotal, such as https://doi.org/10.1126/science.275.5306.1593 ,https://doi.org/10.1038/nature04766; https://doi.org/10.3389/fnins.2012.00157 ) to computational and systems-level accounts of anticipatory reward processing and motivational adaptation across behavioral, neuroimaging, and invasive electrophysiological paradigms are missing. The discussion would certainly benefit from their inclusion.
https://doi.org/10.1093/scan/nsl021 , https://doi.org/10.1016/j.neubiorev.2013.03.023; https://doi.org/10.1002/hbm.22383 https://doi.org/10.3389/fpsyg.2023.1125066; https://doi.org/10.1016/j.jad.2023.02.149 ; https://doi.org/10.1177/2167702620917463 ; https://doi.org/10.1093/cercor/bhn098 ; https://doi.org/10.1073/pnas.1323014111
Author Response
To strengthen the theoretical and neuroscientific framing of the paper and its relation to literature on the neural and behavioral correlates of reward sensitivity and anticipation, I recommend the authors consider integrating some key findings from the literature that are currently missing. While most of the core studies employing the MID task and fMRI meta-analyses are appropriately cited, some important references (some of them pivotal, such as https://doi.org/10.1126/science.275.5306.1593 ,https://doi.org/10.1038/nature04766; https://doi.org/10.3389/fnins.2012.00157) to computational and systems-level accounts of anticipatory reward processing and motivational adaptation across behavioral, neuroimaging, and invasive electrophysiological paradigms are missing. The discussion would certainly benefit from their inclusion.
https://doi.org/10.1093/scan/nsl021 , https://doi.org/10.1016/j.neubiorev.2013.03.023; https://doi.org/10.1002/hbm.22383 https://doi.org/10.3389/fpsyg.2023.1125066; https://doi.org/10.1016/j.jad.2023.02.149 ; https://doi.org/10.1177/2167702620917463 ; https://doi.org/10.1093/cercor/bhn098 ; https://doi.org/10.1073/pnas.1323014111
The reviewer suggests incorporating 11 additional references, which can be grouped into three sets. First, a set very relevant to the current paper, as they also cover outcome anticipation and outcome processing and their relation to stress and depression (Porcelli et al., 2012; Fassett-Carman et al., 2023; Ethridge et al., 2020). We have included these references in the following way:
“Importantly, acute stress can dampen reward sensitivity, reducing striatal and orbito-frontal responses to rewards (Porcelli et al., 2012), which may interfere with learning from positive outcomes.” (page 2)
“During reward anticipation specifically, amygdala and Nacc activation moderated the association between the number of life stressors and depressive symptoms (Fassett-Carman et al., 2023).” (page 2)
“... and psychosocial stress reduces neural reward responsivity (Ethridge et al., 2020).” (page 2)
“Tentatively, our findings of a negative correlation in the amygdala between academic buoyancy and the anticipation of a gain may point to an alternative mechanism. Rather than engaging in stronger emotional responses, academically buoyant students may regulate affective responses during reward anticipation, potentially as a means of maintaining focus and motivation under pressure. This idea fits with evidence that different brain regions contribute distinctively to motivational control: For example, anterior insula activation during reward anticipation is modulated by attention, reflecting conscious evaluation of motivational cues, whereas ventral striatal responses occur automatically and independent of attention (Rothkirch et al., 2014). Given the exploratory nature of this finding and its divergence from prior work, further research is needed to clarify the role of amygdala activity during reward anticipation in the context of academic stressors.” (page 10)
“These results appear to diverge from resilience research showing that stronger amygdala activation during reward anticipation buffered against the impact of life stressors (Fassett-Carman et al., 2023). Other studies similarly highlight that stress reduces neural reward responsibility, while stronger baseline reward responsiveness predicts lower stress reactivity (Pegg & Kujawa, 2023).” (page 10)
The second set of references focuses on outcome processing in reinforcement learning and decision making paradigms (Schultz et al., 1997; Daw et al., 2006; Cohen, 2007; Garrison et al., 2013; Gläscher et al., 2009; Telzer et al., 2014). We included these references on the understanding that we make it explicit that the current paper does not involve reinforcement learning and decision making, but only anticipation and processing of outcomes. We have included these references in the discussion in the following way:
“...and heightened ventral striatal reward reactivity has been associated with decreases in depressive symptoms over time (Telzer et al., 2014)” (page 2)
“These findings are consistent with the reward prediction error model, where dopaminergic neurons code deviations between expected and actual outcomes (Schultz, Dayan, & Montague, 1997), and with reinforcement learning research showing that striatal activity tracks prediction errors across paradigms (Garrison et al., 2013). Reward sensitivity supports adaptive decision-making under uncertainty, shaped by individual differences in reinforcement learning parameters (Daw et al., 2006; Cohen, 2007). Importantly, acute stress can dampen reward sensitivity, reducing striatal and orbitofrontal responses to rewards (Porcelli et al., 2012), and psychosocial stress reduces neural reward responsivity (Ethridge et al., 2020).” (page 2)
The third set of references included a paper on strategic modulation of responses in the stop signal task, which we found difficult to integrate into the current paper (Giuffrida et al., 2023) as our focus is on reward anticipation and academic buoyancy rather than inhibitory control. We also decided not to include Rothkirch and colleagues (2014), as their findings concern attention-dependent modulation of insula and striatal responses, which does not directly relate to our amygdala-focused results on academic buoyancy and would unnecessarily complicate the discussion. We thus did not include these references, but we will do so if the reviewer and editor judge them to be important.
Reviewer 2 Report
Comments and Suggestions for Authors
The authors report a study examining brain responses during incentive anticipation and receipt, and explores the potential association between individual differences in neural reward anticipation responses and academic buoyancy. I found the topic interesting and the manuscript clearly written. There are a few important issues that I believe can be addressed in revision.
- The authors emphasize that this paper aims to contribute to the literature on reward processing and academic achievement by examining a variant of the monetary incentive delay (MID) task and demonstrating similar reward anticipation and receipt responses when the incentives are changed from money to points (framed as hypothetical grade points). Given these goals, I think that the authors must conduct an analysis similarly to typical analysis of the MID task. The current analysis differs on the following points from the typical MID treatment of data (e.g., Knutson et al, 2008: https://doi.org/10.1016/j.biopsych.2007.07.023; Knutson et al., 2005: https://doi.org/10.1523/JNEUROSCI.0642-05.2005):
- The authors chose to model the period following cue offset until the target onset as the anticipatory phase of the trial which was used to estimate reward anticipation signal. This approach fails to acknowledge that participants know what to anticipate as soon as the cue appears (it does not take participants two seconds to recognize the cue – early versions of the MID task used shorter cue durations). This approach also makes the findings harder to compare with other MID task studies, where the typical approach is to model reward anticipation beginning at cue onset. Focusing on the period leading up to the target may mean results are more influenced by motor preparation activity – an analysis of response times may show that response speed is influenced by the anticipation condition and the authors may consider including response times as a motor preparation-related covariate during the cue/anticipation phase as in Knutson et al., 2005: https://doi.org/10.1523/JNEUROSCI.0642-05.2005.
- I was surprised to see that the authors used “individual z values” in the group level permutation test. Contrast estimates (“cope” images rather than “zstat” images in FSL) would be a better measure of response magnitude and is the common approach.
- Head motion is a concern in any speeded response task, but the authors do not report any procedures to minimize effects of head motion on estimated signal. Typically, volume to volume estimates of head motion are included in the first level model and/or individual-level estimates of head motion are included in the group model.
- The authors included a manipulation of incentive magnitude (1/.5 points) but did not examine any magnitude effects in the analysis.
- It appears that Figure 3 is mislabeled – the caption and image labels state that images are of anticipation phase brain signal (same as Figure 2), but the Results text states that Figure 3 describes the receipt phase.
- There are clear a priori regions of interest here and the report could provide useful estimates of effect size for the potential correlation with academic buoyancy by averaging signal over voxels in independently defined regions (defined by atlas or meta-analysis) and reporting the correlation.
- The authors state that participants were asked how much “the task felt like they were achieving a ‘real’ school grade,” but I did not see any results reported from this question. It seems important to the authors’ claims of greater ecological validity (p.10). Without data showing that the “school grade” framing was meaningful to participants, I believe most readers will interpret the incentives in this task as non-specific points, rather than an incentive that is specifically related to academic contexts.
- The introduction reports several findings related to reward sensitivity – some neural measures of reward responses and some self-report measures. It would be helpful if the authors clearly distinguished these findings in the brief review of the literature.
Author Response
- The authors emphasize that this paper aims to contribute to the literature on reward processing and academic achievement by examining a variant of the monetary incentive delay (MID) task and demonstrating similar reward anticipation and receipt responses when the incentives are changed from money to points (framed as hypothetical grade points). Given these goals, I think that the authors must conduct an analysis similarly to typical analysis of the MID task. The current analysis differs on the following points from the typical MID treatment of data (e.g., Knutson et al, 2008: https://doi.org/10.1016/j.biopsych.2007.07.023; Knutson et al., 2005: https://doi.org/10.1523/JNEUROSCI.0642-05.2005):
- The authors chose to model the period following cue offset until the target onset as the anticipatory phase of the trial which was used to estimate reward anticipation signal. This approach fails to acknowledge that participants know what to anticipate as soon as the cue appears (it does not take participants two seconds to recognize the cue – early versions of the MID task used shorter cue durations). This approach also makes the findings harder to compare with other MID task studies, where the typical approach is to model reward anticipation beginning at cue onset.
We acknowledge that our events modeling is different with respect to previous work employing the MID. We adopted this strategy from a large multicenter resilience study the senior author is involved in (DynaMORE; www.dynamore-project.eu), which is itself adopted from another large German resilience study (MARP; https://lir-mainz.de/en/mainzer-resilienz-projekt-marp). In both studies, the same version of the MID was used, which functioned as the template for our school-grade version. We therefore wanted to be maximally compatible with those benchmark studies in the field of mental resilience. Nevertheless, we investigated whether our choice of model impacted the results, but found no meaningful difference.
For example, below is an overlay of the results with cues modelled as separate events in blue on the results with cues included in the anticipation regressors in red for the main effect of anticipation gain > anticipation zero. Purple indicates overlap.
And an overlay of the results with cues modelled as separate events in blue on the results with cues included in the anticipation regressors in red for the association between the ABS scores and the anticipation gain > anticipation zero contrast. Purple indicates overlap.
As the findings are highly similar between our original analyses modeling the cues separately and the ones including the cues in the anticipation period, we hope the reviewer agrees with only reporting our original results based on the cues modeled separately.
- Focusing on the period leading up to the target may mean results are more influenced by motor preparation activity – an analysis of response times may show that response speed is influenced by the anticipation condition and the authors may consider including response times as a motor preparation-related covariate during the cue/anticipation phase as in Knutson et al., 2005: https://doi.org/10.1523/JNEUROSCI.0642-05.2005.
We agree with the reviewer that motor preparation could impact analysis of anticipation-related activation. We therefore looked at the descriptives of the reaction times (RTs) across participants:
And ran a repeated-measures ANOVA with incentive magnitude as within-subject factor:
Although small, this analysis indeed indicated a difference in RTs across the incentive magnitudes (p < .001), with post-hoc differences emerging between 0 and +0.5 and between -0.5 and -1. We therefore reran the analyses, this time using RT on each trial as a weight in the anticipation EVs, as suggested by the reviewer, but found no meaningful difference in results.
For example, below is an overlay of the results with RTs as weights in blue on the results without RTs in red for the main effect of anticipation gain > anticipation zero. Purple indicates overlap.
And an overlay of the results with RTs as weights in blue on the results without RTs in red for the association between the ABS scores and the anticipation gain > anticipation zero contrast. Purple indicates overlap.
As the findings are highly similar between our original analyses and the ones using RTs as weights for the anticipation EVs, we hope the reviewer agrees with only reporting our original results.
While analysing the repeated-measures ANOVA with incentive magnitude as within-subject factor, we noticed that one of our original analyses had a mistake in them. We apologise for the mistake and have corrected it in the results:
“A repeated-measures ANOVA with incentive magnitude as within-subject factor revealed a small but significant effect (p < .001), with post-hoc tests indicating differences between 0 and +0.5 as well as between –0.5 and –1.” (page 7)
- I was surprised to see that the authors used “individual z values” in the group level permutation test. Contrast estimates (“cope” images rather than “zstat” images in FSL) would be a better measure of response magnitude and is the common approach.
The reason we opted for z-values was to take into account within-subject variation (information from the “varcope” image in FSL), thereby resembling a mixed-effects analysis. Using contrast estimates only, it would resemble a random-effects analysis. Nevertheless, we reran all permutation tests using the cope images instead of the z-stat images and found no meaningful difference in results.
For example, below is an overlay of the z-stat results in blue on the cope results in red for the main effect of anticipation gain > anticipation zero. Purple indicates overlap.
And an overlay of the cope results in blue on the z-stat results in red for the association between the ABS scores and the anticipation gain > anticipation zero contrast. Purple indicates overlap.
As the findings are highly similar between our original analyses on the z-stat images and the ones using the cope images, together with the argument of approximating a mixed-effects analysis, we hope the reviewer agrees with only reporting our original results based on the z-stat images.
- Head motion is a concern in any speeded response task, but the authors do not report any procedures to minimize effects of head motion on estimated signal. Typically, volume to volume estimates of head motion are included in the first level model and/or individual-level estimates of head motion are included in the group model.
Indeed, motion parameters in various forms are often used as regressors of no interest, either at the single-subject level or at the group level. To check whether motion may have influenced our results, we reran the analyses adding the 6 rigid body motion parameters to the general linear model at the single subject level and found no meaningful difference in results.
For example, below is an overlay of the results with motion parameters added in blue on the results without motion parameters added in red for the main effect of anticipation gain > anticipation zero. Purple indicates overlap.
And an overlay of the results with motion parameters added in blue on the results without motion parameters added in red for the association between the ABS scores and the anticipation gain > anticipation zero contrast. Purple indicates overlap.
Although the results are highly similar between the results with and without motion parameters added, we follow the reviewer’s suggestion to follow common practice in fMRI analysis to mitigate potential influence of head motion on the results and now report results from the analyses in which the 6 rigid body motion parameters have been added as regressors to the first-level statistical models. The following sentence was added to the Methods:
“Additionally, the six rigid-body motion parameters estimated during preprocessing were added as EVs of no interest to account for head motion in the analysis.”
and the figures in the results section have been adjusted accordingly.
- The authors included a manipulation of incentive magnitude (1/.5 points) but did not examine any magnitude effects in the analysis.
We agree with the reviewer that we should follow up on the incentive magnitude, as we included the distinct magnitudes as manipulation in our task, even though we did not have any hypotheses related to incentive magnitude in relation to the ABS scores. Below are the bar plots of the mean activation (with SEM) for the five incentive magnitudes, plotted for the left and the right Nacc separately.
The plots show a parametric modulation of incentive magnitude in left and right Nacc activation, highly similar to the results shown by Kampa et al. (2018) using the same task with monetary rewards and in line with previous findings from the MID task (Knutson et al., 2001; Carter, MacInnes, Huettel & Adcock, 2009). Although there was an overall effect of incentive magnitude in left and right Nacc activation (both p < .001), post-hoc contrasts did not show a difference between anticipating a high or low reward (both p > .1), or a high or low loss in the left Nacc (p = .153). There was a difference between anticipating a high or low loss in the right Nacc (p = .01). We now report these additional findings in the results:
“To investigate the incentive magnitude manipulation (1 vs 0.5 points), we examined activation across the five incentive magnitudes separately for left and right nucleus accumbens (NAcc). Figure 5 displays the mean activation (with SEM) for each magnitude. A clear parametric modulation of incentive magnitude was observed in both left and right NAcc activation, consistent with previous findings using the MID task (Knutson et al., 2001; Carter, MacInnes, Huettel & Adcock, 2009) and with Kampa and colleagues (2018) employing the same task with monetary rewards.
Post hoc contrasts did not reveal significant differences between anticipation of high versus low reward in either hemisphere (both p > .1), nor between high versus low loss in the left (p = .0153). In the right NAcc, however, activation was greater during anticipation of a high compared to a low loss (p = .01).” (page 7)
- It appears that Figure 3 is mislabeled – the caption and image labels state that images are of anticipation phase brain signal (same as Figure 2), but the Results text states that Figure 3 describes the receipt phase.
We thank the reviewer for noticing our mistake in Figure 3, and have edited the caption and image labels. - There are clear a priori regions of interest here and the report could provide useful estimates of effect size for the potential correlation with academic buoyancy by averaging signal over voxels in independently defined regions (defined by atlas or meta-analysis) and reporting the correlation.
We indeed had a priori regions of interest for the association between academic buoyancy and reward-related activation in the Nacc, vmPFC, and amygdala. We now formalized this by applying separate ROIs for the bilateral Nacc, bilateral amygdala, and the vmPFC in the permutation tests. These ROI masks were independently created using the Harvard-Oxford cortical and subcortical atlases, using the versions with a 25% probability threshold offered in the FSL package. We applied a Bonferroni correction for testing within three separate ROIs, thus using a TFCE-corrected p < .0167 for the ROI analyses. In these new analyses, the previously reported uncorrected association in the right amygdala now survived small-volume correction (see Figure below). This was not found for either the left amygdala, left or right Nacc, or vmPFC. We decided to now only report and discuss the small-volume corrected association with academic buoyancy in the left amygdala instead of the uncorrected whole-brain associations. Note that we do provide the uncorrected statistical maps in Neurovault for all three ROIs.
We choose to refrain from reporting an r-value for this association, as it would offer an inflated effect size that is uninformative for future studies (i.e., “double dipping”; Kriegeskorte et al. (2009), Nature Neuroscience). Instead, we still show the scatter plot to visualize the association.
We now added the small-volume correction to the Methods section:
”Here, we applied a small-volume correction, separately for three regions of interest (ROIs) that were identified as relevant for this brain-behavior association: bilateral Nacc, bilateral amygdala, and ventromedial PFC (vmPFC). The ROI masks were independently created using the Harvard-Oxford cortical and subcortical atlases, using the versions with a 25% probability threshold in FSL. A Bonferroni correction was applied for testing three separate ROIs, thus setting the significance threshold at p < .0167, TFCE-corrected for multiple comparisons.”
and to the Results section:
"We found a negative association between academic buoyancy and activation in the right amygdala when anticipating reward (see Figure 4), which was not found when anticipating loss. No associations with academic buoyancy were found for either contrast within the Nacc or vmPFC ROIs.” (page 9)
and edited the Discussion on the association between ABS and the right amygdala to reflect the small-volume corrected results rather than the uncorrected whole-brain results:
“Beyond validating the SGID, we examined associations between neural activation and academic buoyancy. After applying small-volume correction within our a priori regions of interest (bilateral Nacc, bilateral amygdala, and vmPFC), we found a negative association between academic buoyancy and activation in the right amygdala during reward anticipation. Contrary to our expectations, we did not find an association between academic buoyancy and ventral striatal and vmPFC activation.
These results appear to diverge from resilience research showing that stronger amygdala activation during reward anticipation buffered against the impact of life stressors (Fassett-Carman et al., 2023). Other studies similarly highlight that stress reduces neural reward responsibility, while stronger baseline reward responsiveness predicts lower stress reactivity (Pegg & Kujawa, 2023). Together, these findings support the idea that reward sensitivity plays a protective role under conditions of academic or emotional strain. Tentatively, the negative correlation between academic buoyancy and amygdala activation during reward anticipation may point to an alternative mechanism. Rather than heightened emotional responses, buoyant students may benefit from regulating affective reactivity to maintain focus and motivation under pressure. Given the exploratory nature of this finding and its divergence from prior work, further research is needed to clarify the role of the amygdala in academic stress contexts.” (page 10)
- The authors state that participants were asked how much “the task felt like they were achieving a ‘real’ school grade,” but I did not see any results reported from this question. It seems important to the authors’ claims of greater ecological validity (p.10). Without data showing that the “school grade” framing was meaningful to participants, I believe most readers will interpret the incentives in this task as non-specific points, rather than an incentive that is specifically related to academic contexts.
We thank the reviewer for the comment, and have edited our results section to further reflect that the school grade framing was meaningful with the following text:
“The task also appeared to feel convincing for the majority of participants, with 70% indicating that the school grade paradigm felt ‘real’ and that they aimed to ‘achieve a good grade’.”
- The introduction reports several findings related to reward sensitivity – some neural measures of reward responses and some self-report measures. It would be helpful if the authors clearly distinguished these findings in the brief review of the literature.
Based on this comment, we have rewritten parts of the introduction to distinguish between neural reward responses and self-report measures. Specifically, the new text now reads as follows:
“ One promising factor is reward sensitivity, or the capacity to experience pleasure both in anticipating and receiving reward-related stimuli (Grey & Gelder, 1987). At the self-report and behavioral level, reduced reward sensitivity is impaired in disorders such as posttraumatic stress disorder (Ben-Zion et al., 2022; Elman et al., 2009; Kasparek et al., 2020) and major depressive disorder (Knutson et al., 2008; Pizzagalli et al., 2007). Not surprisingly, anhedonia, the diminished capacity to feel enjoyment or reward satisfaction, is commonly found in depressive disorders (Must et al., 2006). Stressors may play a significant role in the onset of anhedonic symptoms and associated mental disorders (Corral-Frías et al., 2015). Research suggests that individuals with higher reward sensitivity display higher levels of positive affect following stressor exposure (Corral-Frías et al., 2016) and may better maintain their mental health, compared to individuals with lower reward sensitivity (Geschwind et al., 2010). These findings suggest that behavioral reward sensitivity can buffer against the negative effects of stress.
At the neural level, studies converge on the idea that reward processing supports resilience. For instance, stronger nucleus accumbens (Nacc) activation in response to a reward task predicts fewer depressive symptoms following stress exposure (Tashjian & Galván, 2018), and heightened ventral striatal reward reactivity has been associated with decreases in depressive symptoms over time (Telzer et al., 2014). During reward anticipation specifically, amygdala and Nacc activation moderated the association between the number of life stressors and depressive symptoms (Fassett-Carman et al., 2023). These findings are consistent with the reward prediction error model, where dopaminergic neurons code deviations between expected and actual outcomes (Schultz, Dayan, & Montague, 1997), and with reinforcement learning research showing that striatal activity tracks prediction errors across paradigms (Garrison et al., 2013). Reward sensitivity is also reflected in adaptive decision-making under uncertainty, where frontopolar and striatal systems balance exploration and exploitation (Daw et al., 2006). Importantly, acute stress can dampen reward sensitivity, reducing striatal and orbitofrontal responses to rewards (Porcelli et al., 2012), which may interfere with learning from positive outcomes. Together, this work indicates that neural sensitivity to reward anticipation may be particularly important for resilience outcomes (Vidal-Ribas et al., 2019).”

Reviewer 3 Report
Comments and Suggestions for Authors
Dear authors
I am writing to you regarding your manuscript titled "Investigating Neural Reward Sensitivity in the School Grade Incentive Delay Task and its Relation to Academic Buoyancy," which I have had the pleasure of reviewing it.
First, I want to commend you on addressing a highly relevant and interesting topic. The intersection of neural reward sensitivity, incentive delay tasks, and academic buoyancy in young students is an important area of research, and your work offers valuable insights.
I have completed my review and have several comments and suggestions that will help strengthen the manuscript further. My feedback is primarily aimed at enhancing clarity, reinforcing methodological rigour, and ensuring the broadest possible impact of your findings.
The detailed comments are provided below [or "in the attached review document," if applicable]. Key areas for your consideration include:
- Clarification of Terminology: Ensuring consistent and precise definitions for "neural reward sensitivity" and "academic buoyancy" throughout the manuscript, particularly for readers who may be new to one of these fields.
- Methodological Details: Providing a bit more detail on [specific methodological point, e.g., "the specific fMRI paradigm used," "the exact nature of the incentive delay task for Grade 2 students," "how academic buoyancy was measured and validated for this age group"].
- Discussion of Limitations: Expanding on the limitations of the study, perhaps considering [e.g., "the generalizability of findings to other age groups or educational contexts," "potential confounds related to cognitive development at Grade 2"].
- Implications for Practice: Further elaborating on the practical implications of your findings for educators, parents, or intervention strategies aimed at fostering academic buoyancy.
Addressing these points will significantly enhance the manuscript's impact and readability. With these revisions, your valuable research will make an even more substantial contribution to the field.
Thank you for the opportunity to review your work. I look forward to seeing the revised version of your manuscript.
Sincerely,

Meticulous proof reading is required
Overall academic tone is solid, but the manuscript would benefit from professional language editing.
Author Response
- Clarification of Terminology: Ensuring consistent and precise definitions for "neural reward sensitivity" and "academic buoyancy" throughout the manuscript, particularly for readers who may be new to one of these fields.
Based on this comment and the other reviewers’ comments, we have rewritten part of the introduction and discussion to further clarify the definitions of neural reward sensitivity and academic buoyancy.
Neural reward sensitivity: “At the neural level, reward sensitivity refers to the responsiveness of fronto-striatal and limbic circuits, particularly the ventral striatum, nucleus accumbens, and amygdala, during reward anticipation and outcome (Schultz et al., 1997).” (page 2)
Academic buoyancy: “and academic buoyancy, which refers to students’ ability to successfully deal with the more common, everyday academic setbacks and pressures (Martin & Marsh, 2008).” (page 1)
- Methodological Details: Providing a bit more detail on [specific methodological point, e.g., "the specific fMRI paradigm used," "the exact nature of the incentive delay task for Grade 2 students," "how academic buoyancy was measured and validated for this age group"].
The specific fMRI paradigm can be found in the methods section, and academic buoyancy was measured using the Academic Buoyancy Scale (ABS). This is a widely-used, validated questionnaire in our age group of 18-30 (e.g. Martin & Marsh, 2008; Martin, 2013), and thus did not require validation. We are unsure what the reviewer means by “Grade 2 students”, as our sample comprised adult students.
- Discussion of Limitations: Expanding on the limitations of the study, perhaps considering [e.g., "the generalizability of findings to other age groups or educational contexts," "potential confounds related to cognitive development at Grade 2"].
One of the limitations in our study is the age group and the generalizability of findings to other age groups. As we mentioned in our paper, we are currently conducting a study in a different age group of children 10 - 14, which will hopefully clarify the generalizability of findings. Once again, we are not sure what the reviewer means by Grade 2.
- Implications for Practice: Further elaborating on the practical implications of your findings for educators, parents, or intervention strategies aimed at fostering academic buoyancy.
We appreciate the reviewer’s suggestion to elaborate on the practical implications of our findings. While we recognise that our results point to potentially meaningful directions for supporting academic buoyancy, we believe it is premature to propose specific strategies for educators or parents on the basis of this initial work. Instead, we now emphasize in the discussion that there is a larger study being conducted based on this initial pilot study, which could, in turn, inform the design of interventions aimed at fostering academic buoyancy.
Round 2
Reviewer 2 Report
Comments and Suggestions for Authors
The authors gave a well-reasoned reply to issues raised in the original submission. No further comments.